# Synergistic Interactions between Linalool and Some Antimycotic Agents against *Candida* spp. as a Basis for Developing New Antifungal Preparations

Anna Biernasiuk *  and Anna Malm 

Department of Pharmaceutical Microbiology, Faculty of Pharmacy, Medical University of Lublin,
20-093 Lublin, Poland; anna.malm@umlub.pl
* Correspondence: anna.biernasiuk@umlub.pl; Tel.: +48-81448-7100

**Abstract:** The incidence of superficial infections, including oral candidiasis, has recently increased significantly. Their treatment is quite difficult due to the growing resistance of *Candida* spp. to antifungal agents. Therefore, it is necessary to search for novel antimycotics or alternative antifungal therapies. The purpose of the study was to evaluate the antifungal activity of natural terpene—linalool (LIN)—against both reference fungi belonging to yeasts and *Candida* spp. isolates from the oral cavities of immunocompromised, hemato-oncology patients. Moreover, its mechanism of action and interactions with selected antifungal drugs or antiseptics were investigated. The broth microdilution technique, ergosterol or sorbitol tests, and a checkerboard method were used for individual studies. The LIN showed potential activity toward studied strains of fungi with a minimal inhibitory concentration (MIC) in the range of 0.5–8 mg/mL and fungicidal effect. This compound was also found to bind to ergosterol in the yeast cell membrane. Additionally, the interactions between LIN with antiseptics such as chlorhexidine, cetylpyridinium, and triclosan showed beneficial synergistic effect (with FIC = 0.3125–0.375), or an additive effect with silver nitrate and chlorquinaldol (FIC = 0.625–1). Moreover, statistically significant differences in MIC values were found for the synergistic combinations of LIN. No interaction was indicated for nystatin. These results confirm that the LIN seems to be a promising plant component used alone or in combination with other antimycotics in the prevention and treatment of superficial fungal infections. However, further clinical trials are required.

**Keywords:** linalool; synergism; interaction; mode of action; *Candida* spp.

## 1. Introduction

Nowadays, the incidence of fungal infections both community-acquired and nosocomial has significantly raised, especially in patients from high-risk groups (with impaired immunity, patients with HIV/AIDS, organ transplant recipients, cancer patients receiving chemotherapy, individuals undergoing broad-spectrum antibiotic and corticosteroid therapy, or immunosuppression). Among different species of fungi, yeasts belonging to *Candida* are opportunistic microorganisms that colonize healthy individuals [1–4]. They are also the most widespread pathogens responsible for the majority of fungal infections causing diverse clinical diseases, ranging from superficial and mucosal candidiasis to invasive diseases associated with candidemia, metastatic organs, and potentially life-threatening systemic illness. *Candida* is considered as the main cause of nosocomial fungal infections (almost 80%) [4–6]. These fungi are a significant health problem not only for immunocompromised patients, but also for healthy people [4,6]. Over 20 *Candida* species can cause infections [7], but *C. albicans* is responsible for the most number of cases of candidiasis and is associated with high mortality (up to 35–50%). The number of infections caused by other *Candida* spp., such as *C. glabrata*, *C. tropicalis*, or *C. krusei* is also increasing [4,5,7,8].

The treatment of fungal infections is still difficult and insufficient. Only a few groups of drugs are used in the therapy of candidiasis, namely polyenes, echinocandins, triazole derivatives, allylamines, and flucytosine [3–6,8]. Moreover, none of them meet all the expectations. Additionally, the Centers for Disease Control and Prevention (CDC) considered that the growing resistance of *Candida* to antifungals is an increasing public health problem worldwide [7]. It is worth noting that emerging pathogenic yeasts may be resistant to many classes of available antimycotics (e.g., *C. auris*—multidrug-resistant microorganism described in 2009, and already reported from thirty-six countries on six continents, exhibiting variable susceptibility to azoles, echinocandins, and amphotericin B) [8–11].

Recently, plant-derived compounds with multidirectional biological properties have been widely identified and investigated [12]. The secondary plant metabolites may be important sources of new antifungal drugs with anti-*Candida* activity or constituents suitable for further modification [12,13]. According to Zida et al. [13], a total of 111 articles were reported (between 1966 and 2015) in which 142 anti-*C. albicans* phytochemicals were distinguished. Among them, 71 (50%) compounds were active (with MIC values below 8 mg/mL) and 60 (42.25%) is noteworthy (MICs below 1 mg/mL). Moreover, 24 (16.9%) of these natural products can be classified as fungicidal. Additionally, the antifungal activity of 16 (11.27%) substances was confirmed also in gold standard in vivo experiments. In another review, Lu et al. [14] summarized the anti-*Candida* activity of phytochemicals published after 2010 (in 2010–2017), especially those with MICs $\leq$ 32 µg/mL. According to them, multicenter studies showed that certain phytocompounds, such as phenylpropanoids, flavonoids, alkaloids, and terpenoids possess promising antifungal properties toward *Candida* spp. Some of them indicated antifungal effect, with MICs of $\leq$8 µg/mL, and a higher activity against drug-resistant *Candida* spp. than fluconazole or itraconazole [14]. Other authors exhibited that the combinations of fluconazole with different natural compounds or extracts were effective even toward fluconazole-resistant *C. albicans* strains [15].

Phenylopropanoids (i.e., phenylpropanoic acids, coumarins, and lignans) have been studied for anticandidal activities. Some of them (anisylalcohol, salicylaldehyde, chlorogenic, caffeic, and quinic acids from phenylpropanoic acids or scopoletin belonging to coumarins) showed activity against *Candida* spp. at MICs of 8–31 µg/mL. The subsequent phenylpropanoic acids (coniferyl aldehyde, cinnamaldehyde, sinapaldehyde, estragole, eugenol, and methyleugenol) exhibited lower anticandidal efficacy (with MIC values exceeding 100 µg/mL). The neolignans (honokiol, magnolol, or glochidioboside) at MICs of 3.3–25 µg/mL also possessed an inhibitory effect toward *Candida* strains [14].

There are several reports of significant antifungal activity of quinones, such as purpurin, aloeemodin, shikonin, or menadione (MIC = 1.28–15.6 µg/mL). The anticandidal effect also showed flavonoids: baicalein, myricetin, isoquercitrin, quercetin, kaempferol, derrone, and licoflavone C (MICs of 1.9–64 µg/mL). Additionally, the promising effect against *C. albicans* indicated rutin, papyriflavonol A, pinocembrin, genistein, silibinin, and alpinetin (MIC = 6.25–32 µg/mL). Moreover, some alkaloids have strong activity toward *Candida* spp., especially tylophorinine hydrochloride or tylophorinidine hydrochloride, vincamine, trigonelline (MIC = 0.6–8 µg/mL), and tetrandrine (MIC = 32 µg/mL). The anti-*Candida* activity of other alkaloids (berberine, roemerine, or matrine) was also reported [14].

In addition, there are also many studies on the antifungal effect of terpenes and terpenoids. Among them, laurepoxyene and laurokamurene C possessed potent anticandidal activity even at MICs of 1–2 µg/mL and plumericin or its isomers at MICs < 4 µg/mL. Hinokitiol, rubiarbonol G and retigeric acid B inhibited growth of *C. albicans* strains at MIC = 5–16 µg/mL. Isopimaric acid and some diterpenoids (MIC = 15–18 µg/mL), or pseudolaric acid B (MICs of 16–128 µg/mL) were also active against *Candida* spp. In turn, linalool, carvacrol, thymol, citral, geraniol, citronellol, and citronellal were responsible for the anti-*Candida* activity of the essential oils (EOs) from many plants [14].

Therefore, it would be an excellent idea to use EOs and their selected constituents to develop new potential phytopharmaceuticals with antifungal effect. EOs are well-known as

agents with broad-spectrum activity. They are rich mixtures of different substances, including terpenes (monoterpenes, e.g., linalool, thymol, menthol, geraniol, and sesquiterpenes), terpenoids, alcohols, phenols, aldehydes, ketones, esters, ethers, and other components with low molecular weights produced by aromatic plants [5,16]. One type of EO may contain even over 100 various compounds in different ratios (1–70%) with 2–3 main components constituting 20–70% of the total composition [5,14,17,18].

Linalool (LIN), also known as 3,7-dimethyl-1,6-octadien-3-ol ($C_{10}H_{18}O$), is an unsaturated aliphatic alcohol belonging to the terpene group—monoterpene [19–24]. It is a volatile flavor compound which is produced by over 200 plants worldwide (lavender, basil, coriander, jasmine, rosewood, linaloe, rosemary, rose, petitgrain, and bergamot) and isolated from their flowers, leaves, herbs, seeds, and wood [20,22,25]. LIN was approved by the Food and Drug Administration (FDA) as Generally Recognized as Safe (GRAS) [1,19,25–27]. Therefore, it is widely use in various industry sectors, including the food as well as the perfume, cosmetic, and pharmaceutical industries [23–25].

This terpen and EOs rich in LIN exhibit various biological activities, including antimicrobial [1,20,22,24,28,29], anti-inflammatory [1,19,21–24,28], antioxidant [20,21,24], anticancer [20,21,23,28,29], antiplasmodial [19,22], antinociceptive [1,19,22,28], antihyperalgesic, or antihyperlipidemic [19,22]. LIN is also known as an intermediate in the biosynthesis of vitamins A and E [19,20,22,30]. In addition, it prevents complications in diabetes, atherosclerosis of the coronary arteries, Alzheimer's disease, and aging processes [25]. Its anxiolytic, antidepressive, and neuroprotective activities were also demonstrated in several studies [28,29]. Moreover, in oral hygiene and dentistry, it is used as an ingredient of mouthwashes, toothpastes, and antiseptic solutions [17,31]. LIN is also a component of perfumed hygiene products or cleaning agents, i.e., soaps, shampoos, lotions, detergents, insecticides, and pest repellents [19,20,23,27,29,30].

In the present work, we verified in vitro the antifungal activity of LIN against reference fungi belonging to yeast from *Candida* spp., *Cryptococcus* spp., *Geotrichum* spp., and *Saccharomyces* spp. Additionally, its antifungal effect toward *Candida* spp. from hemato-oncology patients was investigated. In the next stage, the mode of antifungal action of LIN on *Candida* spp. cells and its potential interactions with selected antimycotics were evaluated.

## 2. Materials and Methods

### 2.1. Materials

#### 2.1.1. The Studied Compounds

In our studies, linalool (3,7-dimethyl-1,6-octadien-3-ol) (Sigma-Aldrich Chemicals, St. Louis, MO, USA) was used to investigate the antifungal effect toward strains of *Candida* spp., *Cryptococcus* spp., *Geotrichum* spp., and *Saccharomyces* spp. Additionally, selected antimycotics, i.e., antibiotic—nystatin, and antiseptics—chlorhexidine (Sigma-Aldrich Chemicals, St. Louis, MO, USA), chlorquinaldol (5,7-dichloro-8-hydroxy-2-methylquinoline), cetylpyridinium chloride monohydrate (cetylpyridinium), silver nitrate, and triclosan (5-chloro-2-(2,4-dichlorophenoxy)phenol) (Glentham Life Sciences, Corsham, UK) were applied to determine the interactions with LIN. In turn, dimethyl sulfoxide (DMSO) (Pol-Aura, Różnowo, Poland) was used to dissolve all compounds in order to obtain their stock solution.

#### 2.1.2. Microbial Species

The reference fungal strains used in the study included: *Candida albicans* ATCC 2091, *Candida albicans* ATCC 10231, *Candida albicans* ATCC 64124, *Candida glabrata* ATCC 90030, *Candida glabrata* ATCC 15126, *Candida parapsilosis* ATCC 22019, *Candida krusei* ATCC 14243, *Candida kefyr* ATCC 66028, *Candida lusitaniae* ATCC 34449, *Candida tropicalis* ATCC 1369, *Candida auris* CDC B11903, *Cryptococcus neoformans* ATCC 90112, *Cryptococcus neoformans* ATCC 32045, *Cryptococcus gatti* ATCC 56992, *Geotrichum candidum* ATCC 34614, and *Saccharomyces cerevisiae* ATCC 9763.

Additionally, 20 isolates of *C. albicans* and 20 isolates of non-*albicans Candida* spp. (NAC), i.e., *C. glabrata*, *C. tropicalis*, *C. parapsilosis*, *C. famata*, *C. krusei*, *C. lusitaniae*, and *C. guilliermondii* were studied. These fungi were isolated from the oral mucosa of hemato-oncology patients, especially those vulnerable to candidiasis. The Ethical Committee of the Medical University of Lublin approved the study protocol (No. KE-0254/75/2011). The standard diagnostic methods were used to identify these isolates [4].

### 2.2. Methods

#### 2.2.1. In Vitro Antifungal Activity Assay of LIN

To determine the antifungal effect of LIN toward reference *Candida* spp., *Cryptococcus* spp., *Geotrichum* spp., *Saccharomyces* spp. strains, and clinical isolates of *Candida* spp., the broth microdilution technique was applied [31–35]. The study was carried out in accordance with the guidelines of the European Committee on Antimicrobial Susceptibility Testing (EUCAST) [34] and Clinical and Laboratory Standards Institute (CLSI) [35] as described previously [4]. The minimal inhibitory concentration (MIC) of LIN was tested using serial 2-fold dilutions in RPMI 1640 broth with MOPS (3-(N-Morpholino)propanesulfonic acid) (Sigma-Aldrich Chemicals, St. Louis, MO, USA). Fungi were cultured on Sabouraud agar (BioMaxima S.A., Lublin, Poland) at 37 °C for 24 h, and then suspended in 0.85% NaCl (0.5 McFarland scale). Afterward, appropriate suspensions were introduced to wells with serial dilutions of terpene (0.03–16 mg/mL) and incubated (37 °C, 24 h). The MIC values of LIN were assessed spectrophotometrically as its lowest concentration showing inhibition of fungal growth. In turn, the minimal fungicidal concentration (MFC), defined as the lowest concentration of a compound required to kill fungi, was evaluated by transferring the cultures used for MIC determination from each well to Sabouraud agar and by incubating as before. The lowest concentration of LIN without the observed fungal growth was interpreted as fungicidal concentration. Additionally, the fungicidal (MFC/MIC $\leq$ 4) or fungistatic (MFC/MIC > 4) effect of LIN was determined [36]. Moreover, the following values were calculated: $MIC_{50}$, $MIC_{90}$, $MFC_{50}$, and $MFC_{90}$. The $MIC_{50}$ or $MIC_{90}$ represented the MIC value at which $\geq$50% or $\geq$90% of the isolates in a test population were inhibited, respectively. $MIC_{50}$ was equivalent to the median MIC value and $MIC_{90}$, the 90th percentile. A similar calculation was applied to the $MFC_{50}$ and $MFC_{90}$ values [37].

#### 2.2.2. Mode of Antifungal Action of LIN

Sorbitol Assay

In order to determine the influence of LIN on the cell wall of yeasts, the sorbitol assay was carried out according to the procedure presented by other authors [6,38–41] and described by us in a previous report [4]. In this method, sorbitol (Sigma-Aldrich Chemicals, St. Louis, MO, USA) in a final concentration of 0.8 M was added to the Sabouraud dextrose broth (SDB) medium (BioMaxima S.A., Lublin, Poland). The MIC values of LIN and nystatin (as control) using SDB with and without sorbitol toward five selected reference strains of *Candida* spp. (*C. albicans* ATCC 2091, *C. albicans* ATCC 10231, *C. parapsilosis* ATCC 22019, *C. glabrata* ATCC 90030, and *C. krusei* ATCC 14243) were assayed. Serial dilutions of LIN and nystatin were in the range 0.03–32 mg/mL and 0.004–1000 μg/mL, respectively. Subsequently, fungal suspensions were introduced into each well, and the microplates were placed at 37 °C. MIC values were read after 2 and 7 days [6,38–41]. Based on the ability of sorbitol to act as an osmotic protector of yeast cell wall, higher MIC values observed in the medium with sorbitol compared to the medium without sorbitol could indicate that the cell wall would be a possible target for LIN.

Ergosterol Assay

Another study was the evaluation of LIN binding to the fungal membrane sterols using the exogenous ergosterol assay. It was carried out in accordance with the procedure described by other researchers [6,38–42] and presented by us in a previous report [4]. An ergosterol (Sigma-Aldrich Chemicals, St. Louis, MO, USA) solution (10 mg/mL) was

used in this test. The MICs of LIN and nystatin (as control) toward five selected reference strains of *Candida* spp. (*C albicans* ATCC 2091, *C. albicans* ATCC 10231, *C. parapsilosis* ATCC 22019, *C. glabrata* ATCC 90030, and *C. krusei* ATCC 14243) were determined using the broth microdilution techniques in the presence and absence of exogenous ergosterol. In SDS medium with and without ergosterol (at a concentration of 400 µg/mL), serial dilutions of LIN and nystatin in the range 0.03–32 mg/mL and 0.004–1000 µg/mL were performed, respectively. Then, fungal suspensions were added to each well, the plates were incubated at 37 °C for 24 h, and MICs were evaluated. The higher MIC in ergosterol medium compared to the ergosterol-free medium may show that the cell membrane would be a possible target for LIN [6,42].

2.2.3. Determination of LIN Interaction in Combination with Selected Antifungal Agents

To assess the interactions of LIN with selected antifungal compounds, a checkerboard method was used. Several antimycotics were investigated in these studies: nystatin, chlorquinaldol, cetylpyridinium, chlorhexidine, silver nitrate, and triclosan. The LIN and antifungal agents listed above were diluted in the broth in appropriate concentrations (based on their MIC values) ranging from 8 times higher than MIC to 8 times lower than MIC. These compounds were introduced horizontally (LIN) and vertically (individual antimycotics) into the microplate. Then, after adding the *C. albicans* ATCC 10231 inoculum to all wells, the plates were incubated as before [6,32,41]. After determining the MICs of LIN alone and in combinations, the fractional inhibitory concentrations (FICs) and FIC index (FICI, $\Sigma$ FIC) were calculated as: $\Sigma$ FIC = FICA + FICB = (CA/MICA) + (CB/MICB), where MICA and MICB are the MIC$_S$ of compounds A (LIN) and B (studied antifungals) alone, respectively. In turn, CA is the MIC value of compound A in combination with B, and CB—MIC value of compound B in combination with A. Finally, FICI values were interpreted as follows: FICI $\leq$ 0.5 as synergism, FICI between 0.5 and 1 as addition, FICI between 1 and 4 as indifference, and FICI > 4 as antagonism [6,32,41].

2.2.4. Data Analysis

Each test was carried out in triplicate and representative data (mode) were shown. Moreover, a statistical analysis (using the Mann–Whitney U test) was performed to compare the activity of LIN alone and in combination with selected antifungals.

## 3. Results

### 3.1. The Antifungal Activity Assessment of LIN

Our data indicated the promising antifungal effectiveness of LIN toward 16 reference fungal strains from four species: *Candida* spp., *Cryptococcus* spp., *Geotrichum* spp., and *Saccharomyces* spp. Considering the results shown in Table 1, this effect was demonstrated at MIC = 0.5–8 mg/mL. The MFCs were the same or 2–4-fold higher, in the range of 0.5–8 mg/mL. The strains belonging to *Cryptococcus* spp. were found to be the most susceptible to LIN (MIC = MFC = 0.5 mg/mL, MFC/MIC = 1). *Geotrichum candidum* strain showed a similar sensitivity at MIC = 0.5–1 mg/mL, and MFC = 1–2 mg/mL (MFC/MIC = 1–2). In turn, the activity of LIN toward *Candida* spp. and *Saccharomyces* spp. was slightly lower with MICs ranging from 1 to 8 mg/mL and MFCs from 2 to 8 mg/mL (MFC/MIC = 1–4). The antifungal activity of LIN against individual strains was similar. It should be added that studied monoterpene showed a beneficial, fungicidal effect with MFC/MIC = 1–4 toward all reference strains.

**Table 1.** The antifungal activity of LIN and NYS (nystatin as control) expressed as a range of MIC or MFC and MFC/MIC ratio toward the reference strains of yeasts.

| Reference Strains | LIN (mg/mL) | | | NYS (µg/mL) | | |
|---|---|---|---|---|---|---|
| | Range of MIC | Range of MFC | MFC/MIC Ratio | MIC | MFC | MFC/MIC Ratio |
| *Candida albicans* ATCC 10231 | 2–8 | 4–8 | 1–2 | 0.48 | 0.48 | 1 |
| *Candida albicans* ATCC 2091 | 2–4 | 4–8 | 2 | 0.24 | 0.24 | 1 |
| *Candida albicans* ATCC 64124 | 2 | 4 | 2 | 0.98 | 1.95 | 2 |
| *Candida glabrata* ATCC 90030 | 2–8 | 4–8 | 1–2 | 0.24 | 0.48 | 2 |
| *Candida glabrata* ATCC 15126 | 2 | 2 | 1 | 0.24 | 0.24 | 1 |
| *Candida parapsilosis* ATCC 22019 | 1–2 | 2–4 | 2 | 0.24 | 0.48 | 2 |
| *Candida krusei* ATCC 14243 | 1–4 | 4–8 | 2–4 | 0.24 | 0.24 | 1 |
| *Candida kefyr* ATCC 66028 | 4 | 4 | 1 | 0.24 | 0.48 | 2 |
| *Candida lusitaniae* ATCC 34449 | 2 | 4 | 2 | 0.48 | 0.98 | 2 |
| *Candida tropicalis* ATCC 1369 | 1–2 | 2 | 1–2 | 0.24 | 0.48 | 2 |
| *Candida auris* CDC B11903 | 2–4 | 4 | 1–2 | 0.98 | 0.98 | 2 |
| *Cryptococcus neoformans* ATCC 90112 | 0.5 | 0.5 | 1 | 0.12 | 0.24 | 2 |
| *Cryptococcus neoformans* ATCC 32045 | 0.5 | 0.5 | 1 | 0.24 | 0.24 | 1 |
| *Cryptococcus gatti* ATCC 56992 | 0.5 | 0.5 | 1 | 0.24 | 0.24 | 1 |
| *Geotichum candidum* ATCC 34614 | 0.5–1 | 1–2 | 1–2 | 0.98 | 0.98 | 1 |
| *Saccharomyces cerevisiae* ATCC 9763 | 2 | 4 | 2 | 0.24 | 0.24 | 1 |

Our data showed the similar anticandidal efficacy of LIN toward both tested groups of clinical yeasts: *C. albicans* and other than *C. albicans*—NAC from hemato-oncology patients. The results, summarized in Table 2 and Figure 1, presented an antifungal activity of LIN with MIC in the range 0.5–8 mg/mL against all *Candida* spp. isolates. The MFCs were the same or 2–4 times higher than MICs, and their ranges were 1–16 mg/mL. Moreover, the minimum concentrations inhibiting the growth of 50% ($MIC_{50}$) or 90% ($MIC_{90}$) of all strains from both of the studied groups were 2 mg/mL and 8 mg/mL, respectively. $MFC_{50}$ and $MFC_{90}$ values were similar ($MFC_{50}$ = 4 mg/mL and $MFC_{90}$ = 8 mg/mL).

**Table 2.** The antifungal activity of LIN expressed as a range of MIC or MFC and $MIC_{50}/MIC_{90}$ ratio or $MFC_{50}/MFC_{90}$ ratio and MFC/MIC ratio toward clinical isolates of *Candida* spp.

| Clinical Isolates | MIC and MFC Values (mg/mL) | | | | Number (%) of Isolates with MFC/MIC Ratio | | |
|---|---|---|---|---|---|---|---|
| | Range of MIC | Range of MFC | $MIC_{50}/MIC_{90}$ | $MFC_{50}/MFC_{90}$ | 1 | 2 | 4 |
| *C. albicans* | 0.5–8 | 1–16 | 2/8 | 4/8 | 8 (40) | 9 (45) | 3 (15) |
| non-*albicans Candida* spp. | | | | | 2 (10) | 11 (55) | 7 (35) |

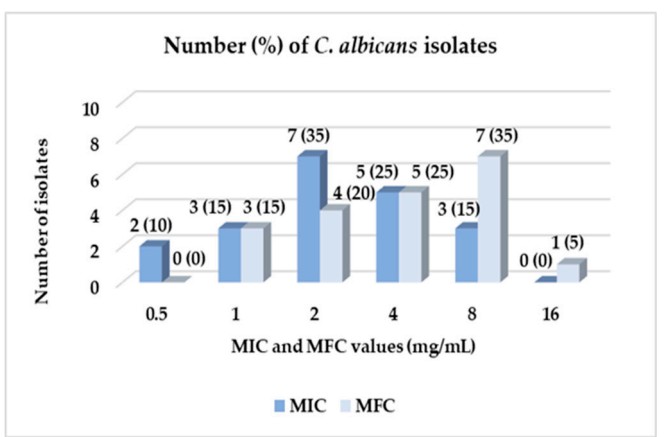 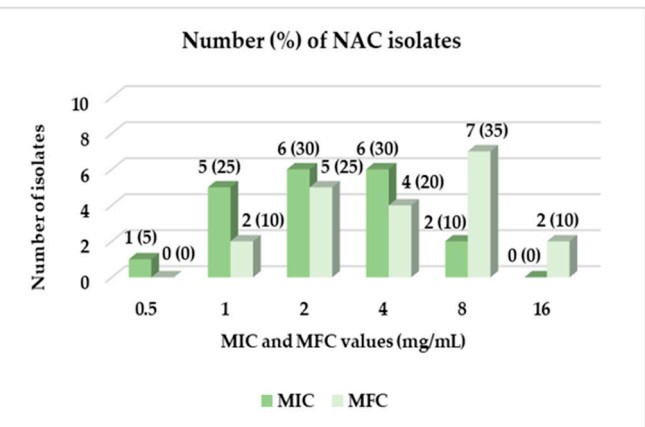

**Figure 1.** Distribution of MIC and MFC values of LIN among clinical isolates of *Candida* spp.

The results also showed that the largest number of studied *Candida* isolates was inhibited by LIN at MIC = 2–4 mg/mL (5 (25%)–7 (35%) strains). Based on MFCs, most of the isolates (7 each (35%) in both tested groups of yeasts) were killed at a minimal concentration of 8 mg/mL. Only single strains (1 (5%)–2 (10%) isolates) were especially sensitive to LIN at MIC = 0.5 mg/mL (Figure 1).

Additionally, the evaluation of both MIC and MFC values of terpene allowed for the calculation of its MFC/MIC ratio (Table 2). In the case of all isolates of *Candida* spp., the fungicidal effect (at MFC/MIC ≤ 4) of LIN was observed. Fungistatic activity (at MFC/MIC > 4) was not found. Analyzing these data, it was shown that the most common MFC/MIC ratio was 2 with a frequency of 45% for *C. albicans* and 55% for NAC isolates.

The obtained results confirmed that LIN had a beneficial fungicidal activity against *Candida* spp. isolated from the oral cavities of clinical patients.

### 3.2. Mechanism of Antifungal Action of LIN

The mode of action of LIN was investigated to assess whether its antifungal effect is due to interaction with the cell wall structure of *Candida* spp. (via the sorbitol test) and/or with the ion permeability of the organism's membrane (via the ergosterol test).

In the sorbitol test, MIC determinations were carried out with and without 0.8 M sorbitol, a known osmoprotectant. The MIC of a cell wall-damaging compound is expected to increase in the presence of this compound. Our results, shown in Figure 2, indicated that the MIC value of LIN did not change in the sorbitol medium after the 7-day incubation. This suggests that LIN most likely does not exert its antifungal effect at the level of cell wall. The MIC value of nystatin (negative control), which acts on the fungal cell membrane, remained the same in the presence of sorbitol.

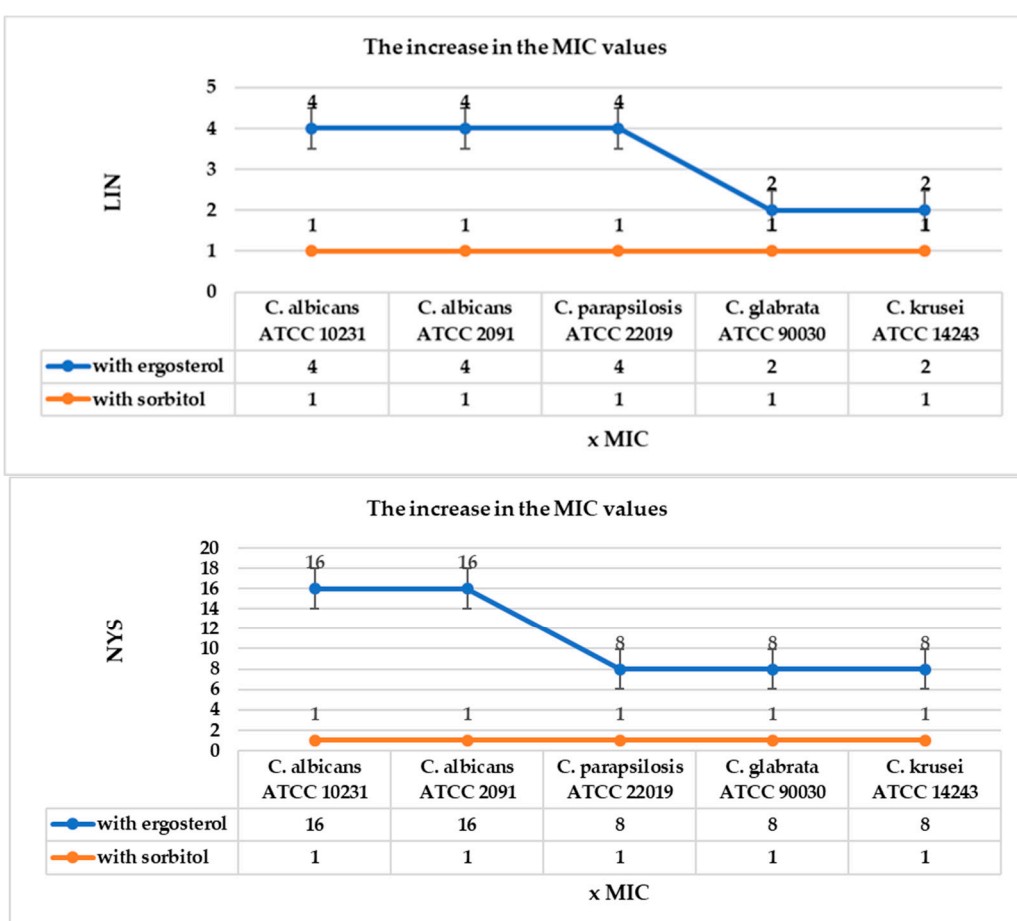

**Figure 2.** The increase in the MIC values (x MIC) of LIN and NYS in the presence of ergosterol and sorbitol toward selected reference *Candida* spp. strains.

The ergosterol assay is based on the binding of exogenous ergosterol to the tested compound, thus preventing its complexation with ergosterol within the cell membrane. The result is an increase in MIC value. Our data, presented in Figure 2, showed 2–4-fold increased MIC values of LIN in the medium with ergosterol compared to those without ergosterol. Similarly, MIC values of nystatin, a known antifungal agent acting via membrane ergosterol binding (positive control), exhibited an 8–16-fold increase. The obtained results suggest that the mode of LIN action may be related with binding to ergosterol in the membrane, leading to an increase in permeability and cell death.

### 3.3. Evaluation of Interaction of LIN with Selected Antifungal Agents

In the next step of our studies, the effect of the combination of terpene with nystatin and some antiseptics—chlorquinaldol, cetylpyridinium, chlorhexidine, silver nitrate and triclosan—on the growth of one of the reference strains, i.e., *C. albicans* ATCC 10231, was evaluated. *C. albicans* is most often isolated from candidiasis, so it was used for further research. The potential interactions of LIN with antifungals listed above were determined using the checkerboard technique. MICs of LIN and studied antimycotics alone, as well as their MICs in combinations, were used to calculate fractional inhibitory concentration (FIC) and FIC index (FICI, Σ FIC) values, which were then accordingly interpreted [32].

The data shown in Table 3 and Figure 3 present three different types of interactions between LIN and studied antifungals: synergy, addition, and indifference toward *C. albicans*. Notably, no antagonism was found. The MIC value of LIN alone was 8000 µg/mL, while in the combination, its MICs were reduced to 2–16-fold depending on the antimycotic. The MICs of the antifungals in the combination also decreased.

**Table 3.** The effect of the combination of LIN with selected antimycotics against *C. albicans* ATCC 10231.

| Antifungal Agent | MIC of Antifungal Agent (µg/mL) | | *p* * | FIC | Σ FIC (FICI) | Interpretation |
|---|---|---|---|---|---|---|
| | Alone | Combination | | | | |
| LIN nystatin | 8000 0.48 | 500 0.48 | 0.012 0.917 | 0.0625 1 | 1.0625 | indifference |
| LIN cetylpyridinium | 8000 3.91 | 1000 0.98 | 0.016 0.012 | 0.125 0.25 | 0.375 | synergism |
| LIN chlorhexidine | 8000 7.81 | 500 1.95 | 0.012 0.012 | 0.0625 0.25 | 0.3125 | synergism |
| LIN chlorquinaldol | 8000 0.98 | 4000 0.48 | 0.175 0.060 | 0.5 0.5 | 1 | addition |
| LIN silver nitrate | 8000 7.81 | 1000 3.91 | 0.016 0.060 | 0.125 0.5 | 0.625 | addition |
| LIN triclosan | 8000 7.81 | 1000 1.95 | 0.012 0.012 | 0.125 0.25 | 0.375 | synergism |

* Mann–Whitney U test.

Combining LIN with some antiseptics was a very good idea. LIN showed beneficial synergistic interactions with cetylpyridinium, chlorhexidine, and triclosan against studied strains. FICI values were in the range 0.3125–0.375. The most favorable combination was shown for terpene with chlorhexidine (FICI = 0.3125). The MIC values of LIN and this antiseptic alone were reduced even 16 and 4 times in combination, from 8000 µg/mL to 500 µg/mL (FIC = 0.0625) and from 7.81 µg/mL to 1.95 µg/mL (FIC = 0.25), respectively (Table 3, Figure 3b).

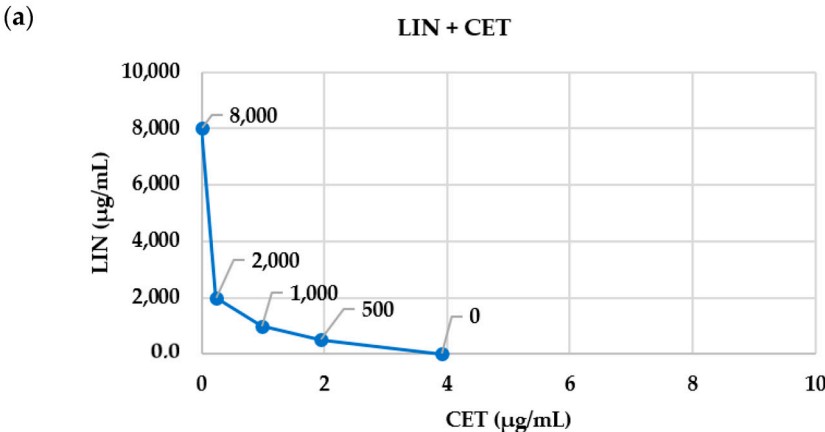

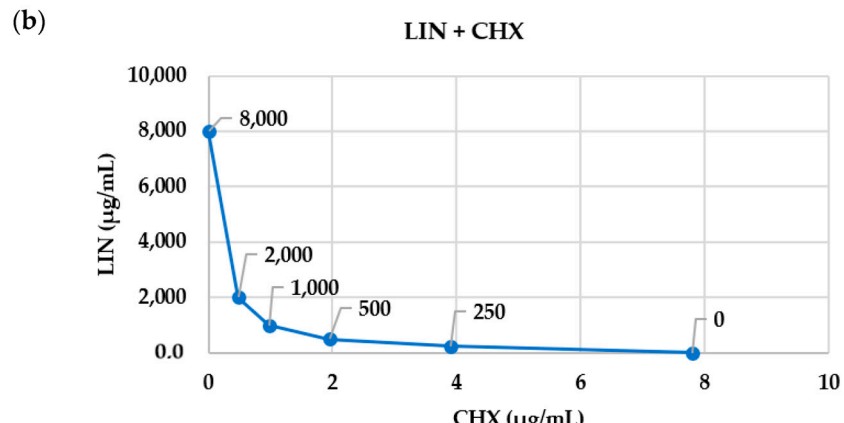

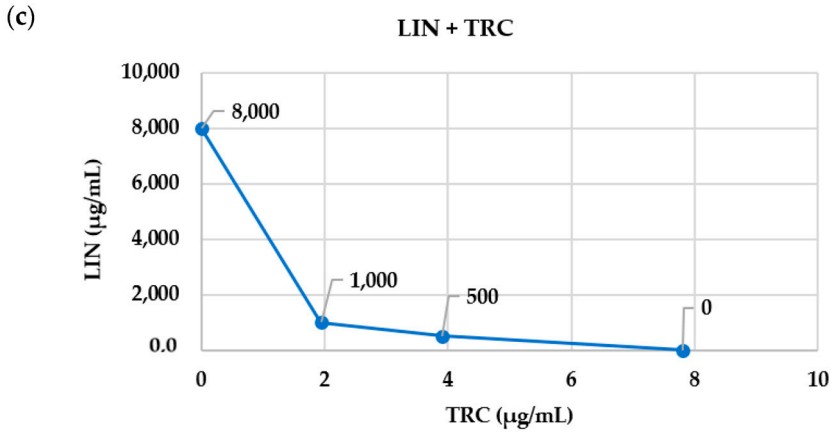

**Figure 3.** Isobolograms indicating the synergy of linalool (LIN) with (**a**) cetylpyridinium (CET), (**b**) chlorhexidine (CHX), and (**c**) triclosan (TRC) for *C. albicans* ATCC 10231.

In the case of the combination of LIN with cetylpyridinium or triclosan, synergism at FICI = 0.375 was observed. The MIC of terpene was also reduced 8-fold in the combination with both of the antiseptics compared to its MIC alone (from 8000 µg/mL to 1000 µg/mL, FIC = 0.125). In turn, MICs of cetylpyridinium and triclosan decreased 4 times, from 3.91 µg/mL to 0.98 µg/mL and from 7.81 µg/mL to 1.95 µg/mL, respectively. The FIC values were the same—0.25 (Table 3, Figure 3a,c).

Moreover, an addition interaction of LIN in combination with chlorquinaldol (FICI = 1) and silver nitrate (FICI = 0.625) was indicated. *C. albicans* strains were 2- and 8-fold

more sensitive to LIN in combination with these antifungals, respectively. MIC of LIN was reduced from 8000 µg/mL to 4000 µg/mL with chlorquinaldol (FIC = 0.5) and to 1000 µg/mL with silver nitrate (FIC = 0.125). The MIC values of both antiseptics were also reduced 2-fold in combination with LIN compared to their MICs alone: from 0.98 µg/mL to 0.48 µg/mL and from 7.81 µg/mL to 3.91 µg/mL, respectively (Table 3).

The indifference effect was demonstrated only when terpen was combined with nystatin (FICI = 1.0625). The MIC value of LIN alone and its MIC with antibiotic differed by 16-fold (first MIC was 8000 µg/mL, and then decreased to 500 µg/mL, FIC = 0.0625), while nystatin MICs were identical (MICs = 0.48 µg/mL and FIC = 1) (Table 3).

Moreover, the Mann–Whitney U test (Table 3) indicated statistically significant differences in MIC values for the synergistic combinations LIN and cetylpyridinium, LIN and chlorhexidine, and LIN and triclosan ($p = 0.016$ or $p = 0.012$, respectively).

The obtained results showed the satisfactory effect of combining LIN with all tested antimycotics, especially with chlorhexidine, cetylpyridinium, and triclosan.

The analysis of the isobologram presented in Figure 3, which was used to assess the interactions between them, also confirmed this beneficial effect.

## 4. Discussion

### 4.1. The Antifungal Activity of LIN

The incidence of fungal infections, especially those caused by *Candida* spp., has recently been steadily increasing, mainly in immunocompromised persons. Their treatment is often inefficient due to the growing resistance of fungi to standard antifungals. This fact is a challenge for clinicians and researchers dealing with the treatment of candidiasis [43] and emphasizes the need to search for new alternative therapies, e.g., from natural sources in combination with other strategies [7,10]. Therefore, we decided to verify the antifungal potential of LIN against 16 reference fungal strains from four species: *Candida* spp., *Cryptococcus* spp., *Geotrichum* spp., and *Saccharomyces* spp., which varied in the range of MIC = MFC = 0.5–8 mg/mL. The ratios of MFC/MIC = 1–4 suggested a beneficial fungicidal effect of terpene on these yeasts. In the case of *Candida* spp., LIN showed activity at MIC = 1–8 mg/mL and MFC = 2–8 mg/mL. The susceptibility of 40 oral isolates from patients with hematological malignancies was similar (MIC = 0.5–8 mg/mL, and MFC = 1–16 mg/mL). Most of the tested *Candida* isolates were inhibited by LIN at MIC = 2–4 mg/mL and only single strains were susceptible to terpene at MIC = 0.5 mg/mL. On the basis of the MFC/MIC ratios, its fungicidal activity was also confirmed.

Similar data were presented by Dias et al. [44]. They analyzed in vitro the antifungal effect of LIN toward isolates of *Candida* spp. from 12 patients with clinical diagnoses of oral candidiasis caused by the use of dentures. In addition, half of these persons reported having diabetes, and five persons reported having inadequate hygiene of oral cavity. MICs and MFCs of LIN were within the range 0.5–2 mg/mL against *C. albicans*, *C. tropicalis*, and *C. krusei*. LIN exhibited fungicidal effect with MFC/MIC = 1–2. The best activity of this terpene was observed toward *C. tropicalis* (MIC = MFC = 0.5 mg/mL), followed by *C. albicans* (MIC = 1–2 mg/mL, MFC = 2 mg/mL) and *C. krusei* (MIC = MFC = 2 mg/mL). Moreover, MICs and MFCs of nystatin, as in our research, were significantly lower (in the range 0.39–0.78 µg/mL). Pandurang et al. [9] also exhibited interesting results with MIC of LIN ranged from 1 to 2 mg/mL against *C. albicans* ATCC 90028 strain and clinical *C. albicans* isolates. In the next studies performed by Cardoso et al. [16], MIC values of this terpene were 0.79–1.58 mg/mL toward two clinical isolates of *C. albicans*. Anticandidal activity of LIN was also tested by Serra et al. [45] and results were similar to that of ours, with MIC = 0.9 mg/mL and MFC = 2.6 mg/mL against both reference and clinical *C. albicans* strains. Moreover, other data indicated activity of LIN toward *C. albicans* with MIC of 1.6 mg/mL [46] or MIC = 0.75 mg/mL [47]. In turn, a slightly higher activity of this terpene was found by further authors. The results obtained by Medeiros et al. [26] showed that 12 strains of *C. albicans* (of which 64.28% included fluconazole-resistant isolates) were sensitive to LIN at MIC = 64–128 µg/mL and MFC = 128–256 µg/mL with MFC/MIC = 1–4 and

fungicidal effect. The susceptibility to LIN of reference strains *C. albicans* ATCC 76485 and *C. albicans* SC 5314 ATCC MYA-2876 was similar (MIC = 64 µg/mL, and MFC = 128 µg/mL). In turn, the activity of nystatin was in the ranges MIC = 4–8 µg/mL and MFC = 4–32 µg/mL. In accordance with other studies [28], clinical *C. krusei* isolates were inhibited by LIN at MICs between 100–200 µg/mL. Its MFC values were 1–2 times greater than the respective MIC values, suggesting that studied terpene showed fungicidal activity. Further data [48] exhibited that LIN inhibited the growth of *C. tropicalis* ATCC 13803 at a concentration of 125 µg/mL. Subsequent results for LIN by El-Sakhawy et al. [49] varied significantly; its mean of MIC value against *C. albicans* isolated from human cutaneous candidiasis was 13.52 µg/mL.

Anti-*Candida* potential of this terpene was also assessed in a study by Zore et al. [50] against thirty-nine isolates of *C. albicans* and nine isolates other than *C. albicans* that were differentially susceptible to fluconazole. LIN inhibited their growth at a concentration of ≤0.064% (*v/v*), showing fungicidal activity. Other results confirmed the antifungal effect of LIN toward 50 clinical isolates of *C. albicans* (28 oropharyngeal and 22 vaginal strains) and *C. albicans* ATCC 3153 at mean MIC = 0.29% and MFC = 0.3% (oropharyngeal isolates), MIC = 0.09%, and MFC = 0.1% (vaginal isolates) [51]. According to further data, MICs of LIN ranged from 0.06 to 0.25% (*v/v*) against 32 different clinical non-*albicans Candida* spp. isolates (*C. krusei, C. parapsilosis, C. lusitaniae, C. norvegensis,* and *C. valida)* from various specimens (blood, body fluids, deep tissue, respiratory tract, gastrointestinal tract, and genital tract of hospitalized patients in Turin (Italy)) [8]. The subsequent studies presented by Marcos-Arias et al. [31] were consistent with the above results and indicated that the range of MICs of LIN was 0.03–0.25% against *Candida* spp. strains (both reference and oral clinical isolates from denture wearers) with $MIC_{50}$ = 0.06% (*v/v*), $MIC_{90}$ = 0.025%, and $MFC_{90}$ = 0.5%.

Hsu et al. [1] showed that LIN at minimal concentrations of 8 mM to 32 mM inhibited the growth of *C. albicans* ATCC 14053 and 18 clinical isolates from blood samples of hospitalized patients (*C. albicans, C. tropicalis,* and *C. glabrata*). The anticandidal activity of LIN was similar with MIC = 16 mM for *C. tropicalis* and *C. glabrata* isolates, and 16–32 mM for *C. albicans*. The MFC values were equal to or twice the MICs (MFC = 16–32 mM), which suggest the fungicidal activity of terpene. In the case of *C. albicans* ATCC 14053, the activity of LIN was slightly stronger (MIC = MFC = 8 mM). Moreover, its sub-MIC concentrations inhibited the formation of germ tubes and biofilms of this strain. Antifungal potential of LIN (MIC = 38.9 µM) towards *C. albicans* was also evaluated in the next data [19]. Additionally, the sensitivity of *C. albicans, C. tropicalis,* and *C. glabrata* strains to this terpene was studied by other authors, and according to them, its MFC values were the same at 3.12 µL/mL against these fungi [7]. The subsequent data [23] indicated the antifungal effect of LIN towards different oral *Candida* isolates from patients with dental problems.

It should be added that our results also demonstrated the potential activity of LIN against *C. auris* CDC B11903 at MIC = 2–4 mg/mL and MFC = 4 mg/mL. This strain presents a new epidemiological problem with high mortality. In the available data, there are not many studies evaluating its susceptibility to LIN. Therefore, our research was fully justified [10,11].

Moreover, our data confirmed some antifungal effect of LIN toward yeasts other than *Candida*. The strains belonging to *Cryptococcus* spp. (*C. neoformans* and *C. gatti*) were the most susceptible to terpen (MIC = MFC = 0.5 mg/mL, MFC/MIC = 1). *G. candidum* showed a similar sensitivity with MIC in the range 0.5–1 mg/mL, and MFC = 1–2 mg/mL (MFC/MIC = 1–2). In accordance with other studies against clinical *C. neoformans* isolates, the MIC value of LIN was 0.79 mg/mL [16] or ranged from 0.5% to 1% [8]. Our results also showed that the activity of LIN toward *S. cerevisiae* with MIC = 2 mg/mL and MFC was two times greater than the MIC. In turn, data presented by other researchers against *S. cerevisiae* exhibited that LIN was effective at MIC of 50 µL/mL [47] and a much lower value of 3.12 µL/mL [7], or MIC in the range 0.12–0.25% [8].

*4.2. Mode of Antifungal Action of LIN*

The next stage of our studies was the evaluation the mechanism of action of LIN on the cell of *Candida* spp. We tried to assess whether its anticandidal activity is related to interaction with the structure of the cell wall and/or cell membrane of yeasts [26,38–41,52]. First, the effect of the LIN on their cell wall was studied. This structure is very specific; it protects the fungal cell from environmental stresses and allows it to interact with its environment. The cell wall and its synthesis is characteristic of fungi, which makes them a very important target for antifungal drugs. Sorbitol, on the other hand, is a factor that induces a certain degree of cellular stress that can cause inhibition of cell growth in the presence of different cell wall inhibitors [5]. It stabilizes fungal protoplasts, which protects their cell wall against external factors. If a compound damages the cell wall, its MIC value in the presence of an osmotic medium increases [38,39,41]. This assay showed that the MICs of LIN did not vary in the presence of sorbitol and were independent of its presence in the medium for any of the five tested *Candida* spp. strains. It suggests that terpene does not act by inhibiting the mechanisms that control cell wall synthesis, but rather by affecting other targets. Nystatin had a similar effect. In the case of both LIN and nystatin, the presence of sorbitol did not interfere with their anticandidal activity. Their MIC values were unchanged, indicating no effect on the structure of the cell wall.

The results are consistent with data obtained for citral and geraniol (also terpenes), which indicated antifungal potential but demonstrated no effect on cell walls [38]. Activity of LIN on the cell wall of *C. albicans* ATCC 76485 and clinical *C. albicans* isolate was also studied by Medeiros et al. [26]. Their results were different compared to ours. The MIC value of LIN for both fungi increased from 64 µg/mL to >1024 µg/mL in medium with sorbitol, indicating that terpene interferes with cell viability through certain molecular mechanisms that may involve the *C. albicans* cell wall and compromise its integrity.

Subsequent studies involved evaluating whether LIN affects ergosterol in the cell membrane. Cell membrane integrity is responsible for cell function [53]. Ergosterol and enzymes involved in its biosynthesis are important targets for some antifungals, such as polyenes or azoles [38,41]. This sterol maintains cell function, and its integrity is involved in cell division, cell signaling, the regulation of membrane proteins, membrane fluidity, and endocytosis. The basis of this study was that the addition of exogenous ergosterol to the medium can increase the MIC value for compounds targeting this sterol in the cell membrane [5,54]. Our results indicated that the cell membrane of studied yeasts may be more sensitive to the action of LIN, as its MIC value was 2–4 times higher in the medium with ergosterol compared to the medium without sterol. Slightly higher MIC increases were found for *C. albicans* compared to non-*albicans Candida* spp. The increase in the MIC of LIN in the presence of ergosterol indicated that this terpen may affect the plasma membrane integrity of *Candida*. In the case of nystatin (binding ergosterol found in lipid bilayer membranes), an 8–16-fold increase in MIC values was indicated. Similar results for nystatin were observed by Castro and Lima [41], in whom the MIC value of the antibiotic against *C. albicans* increased 16-fold in the presence of the sterol.

The obtained data suggest that LIN may affect the fungal cell membrane and bind to this sterol in the membrane, leading to cell membrane disintegration, increasing the permeability of $Ca^{2+}$ and $K^+$ ions, proteins or radicals, and potentially leading to cell death [5,6,38,39,52]. Medeiros et al. [26] evaluated the effect of LIN on the cell membrane of *C. albicans*. Their results were consistent with ours and showed that MIC value of terpene also increased in the medium with exogenous ergosterol. This was an increase in the MIC from 64 µg/mL to >1024 µg/mL. Thus, it was confirmed that LIN affects membrane ergosterol. However, the mechanisms of these interactions are not yet fully understood. According to our results, LIN has a fungicidal effect and probably acts on cell membrane ergosterol, but the exact mechanisms of its action are not known by us. However, there are some reports about the effect of this terpene on the fungus cell. In order to investigate its mechanisms of action, some authors [16] analyzed the inhibition of ergosterol synthesis. The treatment of *C. albicans* and *C. neoformans* strains with LIN at a concentration of

395 µg/mL resulted in a 38% and 57% inhibition of ergosterol synthesis, respectively. In the case of *C. neoformans*, even in subinhibitory concentrations, LIN reduced sterol content. This indicates that it may act in the ergosterol biosynthesis pathway. However, in another study [33], this terpen was not effective in reducing the ergosterol content in *C. albicans*, even at a higher concentration (8 mg/mL). Moreover, molecular docking of LIN with proteins necessary for the biosynthesis and maintenance of the cell wall and the integrity of the fungal plasma membrane showed the possibility of the interaction of this terpene with three important enzymes: 1,3-β-glucan synthase, lanosterol 14-α-demethylase, and D-14-sterol reductase [26].

Recently, LIN was found to exhibit antifungal activity by arresting the *C. albicans* cell cycle. In addition, it induces a reduction in cell size and abnormal germination and inhibition of *C. albicans* germ tube formation (which plays an important role in the virulence) [1,55,56]. LIN also displays anticandidal activity toward the cells in biofilms [1,51]. Some studies confirmed that LIN interfered with the initial stages of *Candida* biofilm formation and was also effective against the artificial biofilms of *C. albicans* [16]. Its antibiofilm potential should be further explored as a therapeutic strategy against biofilm-associated *C. albicans* infections [57]. It should be added that due to the multi-drug resistance exhibited by *Candida* biofilms, biofilm cells are able to survive up to 1000-fold higher concentrations of antifungal agents than those inhibiting non-biofilm planktonic cells [58,59].

Subsequent results exhibited that LIN at a concentration of 0.5% killed 100% of the *C. albicans* ATCC 3153 cells within 30 s. At lower concentrations (even at 0.016%), it inhibits germ tube formation and hyphal elongation of *C. albicans*, showing that it is active toward fungal dimorphism. As a result, it can reduce the progression of fungi and the spread of infection in host tissues [51]. In addition, the exposure of cells to LIN induced decrements in the colony-forming ability, a decrease in the levels of superoxide anion radical and total reactive oxygen species (ROS), and increases in the concentration of peroxides and lipid peroxides, showing oxidative stress induction. LIN treatments resulted also in different adaptive modifications of the antioxidant system [56]. Moreover, LIN at a concentration of 125 µg/mL significantly increased the chromosomal damage of *Candida* spp. [60].

### 4.3. Investigation of Interaction of LIN with Selected Antifungal Agents

The anti-*Candida* activity of LIN may also be exploited using new therapeutic strategies such as combination approach, which is an effective trend in fighting invasive fungal infections [59]. It takes different beneficial characteristics associated with each combined product by improving the efficacy of the individual components. Moreover, in the context of a multi-ingredient combination (of two or more compounds), active compounds may have various mechanisms of action and target sites in the fungal cells. This may enhance the desired antifungal effect, most likely by synergistic and/or additive interactions [7,45,59]. In the case of synergism (synergy), the combination of these substances is much more effective than each compound separately [45]. The combined preparation may also have lower toxicity and fewer side effects due to the reduced concentration of individual ingredients [61]. An appropriate mixing of antimycotics can provide a broader antifungal effect and potentially reduce the risk of resistance in fungi [62,63].

The strategy involving the combination of plant component with conventional antifungals has given encouraging results in recent times. Hence, the antifungal potential of LIN in combination with six different antimycotics was further analyzed by us for the possible synergistic interactions. These studies were carried out in vitro toward *C. albicans* ATCC 10231. *C. albicans* is most often isolated from superficial infections, among others from skin or mucous membrane, oral candidiasis, and also endodontic, periodontal or peri-implant infections [64]. Therefore, this species was used for our study. As for the clinical use of LIN, it is considered as an agent applied in oral hygiene and dentistry (as component of mouthwashes). It is also suggested that its combination with certain antimycotics used in oral candidiasis may enhance the efficacy of both. In addition, for superficial candidiasis, a

topical antifungal agent is preferred for better activity due to a greater penetration of the drug and to avoid systemic side effects [64].

In these studies, we assessed the interactions between LIN and a selected antibiotic, i.e., nystatin, and some antiseptics—cetylpyridinium, chlorquinaldol, chlorhexidine, silver nitrate and triclosan—to evaluate their potential synergistic effect with each other. These compounds are used as antimicrobial agents both in the prevention or treatment of surface and oral infections and in professional oral hygiene. They are components of many oral care preparations, such as mouthwashes and dentifrices, as well as dental hygiene products [4,45,61,65–67].

Our results indicated a promising effect of the combination of LIN with the above antimycotics and confirmed three types of interactions, namely synergism, addition, and indifference. It should be noted that the combined concentrations of the two main components needed to eliminate *C. albicans* were lower. The MIC value of LIN alone was 8000 µg/mL, and in the combination, the value decreased 2–16-fold depending on the antimycotic. Similarly, the MICs of antifungals decreased from 2- to 16-fold (except nystatin). The combinations involving LIN with cetylpyridinium, chlorhexidine, or triclosan were a good idea due to the lowest FICIs (0.3125 or 0.375) and synergistic interactions. Of these, the most favorable combination was shown for terpene with chlorhexidine (FICI = 0.3125). Their MICs were reduced even 4–16 times. The synergistic effect can be related to different target sites and mechanisms of action, which enhances their anticandidal effect. The mode of activity of LIN was previously described in detail (the interaction with the ergosterol in cell membrane). Chlorhexidine, on the other hand, can bind to proteins in the cell wall and lead to a loss of cell integrity, followed by a leakage of cellular constituents [65]. These results suggested that the combined activity of LIN and chlorhexidine toward *C. albicans* was higher than their action alone. The reduced concentration of LIN and chlorhexidine (required to treat infections) in such a mixture will reduce their side effects (especially chlorhexidine, i.e., tooth discoloration, burning sensation or bitter taste) [65].

Additionally, the combination of LIN with either cetylpyridinium or triclosan increased their anticandidal efficacy 4–8 times with identical FICIs of 0.375. Most likely, the mechanism of action of cetylpyridinium can also increase LIN activity. This compound affects the cell by disturbing its osmoregulation and homeostasis. As a result, there may be a leakage of $K^+$ and pentose from the cells as well as autolysis through the activation of intracellular latent ribonucleases. There is also a breakdown of membranes, the leakage of the contents from cytoplasm, damage to nucleic acids and proteins, and the lysis of the cell wall by autolytic enzymes [66]. In the case of triclosan, $K^+$ leakage occurs, which indicates membrane damage, inhibition of ATP-ase enzyme activity, membrane destabilization, impaired ion transport, and the modulation of overall cellular osmoregulation [42]. This may enhance the anticandidal activity of LIN when combined with triclosan.

Moreover, the additive interaction between LIN and silver nitrate (FICI = 0.625) or chlorquinaldol (FICI = 1) was observed. *C. albicans* strains were 8- and 2-fold more susceptible to LIN in combination with these antiseptics, respectively. Single studies report that silver nitrate exhibits antifungal activity probably by destroying membrane integrity. However, the more detailed mechanisms of the antifungal action of this antiseptic still need to be determined [67]. In the case of chlorquinaldol, it is presumed that it uses its lipophilicity to penetrate cell membranes, where its antimicrobial activity is probably related to its chelating activity [68]. Perhaps these properties of silver nitrate and chlorquinaldol also enhanced the antifungal effect of LIN.

In turn, the LIN–nystatin combination indicated an indifference interaction by showing an FICI of 1. This is also a beneficial action where both components can be combined. However, it would not be advisable to lower their concentration. Both LIN and nystatin can affect the cell in a similar way. Nystatin binds to ergosterol in the membrane, causing changes in membrane permeability that allow the release of $K^+$, metabolites, and sugars. It is believed that cell membrane damage is responsible for fungal death [69].

There are no studies in the literature investigating the interaction of LIN with these antifungal agents. Our previous data [4] on the effect of the combination of eugenol (EUG) and clove essential oil (CEO) with selected antimycotics showed their good effect toward four reference *Candida* spp. strains *(C. albicans, C. krusei, C. glabrata,* and *C. parapsilosis).* It was these results that encouraged us to continue research on LIN. Both CEO and EUG indicated synergy with cetylpyridinium. Their MIC values after the combination decreased 4–8 fold. A similar post-combination activity was also found for them with chlorhexidine (synergy and addition). When these component were combined with silver nitrate or triclosan, a 2–8-fold reduction in their MICs and synergism or addition against *Candida* spp. were also observed. In all cases, synergism with a similar FICI = 0.375–0.5 was shown. The indifference, as in these studies, was found only in combination with nystatin (FICI = 1.25–1.5).

However, few reports present the results of the interaction of LIN with other compounds. Particularly noteworthy are the results presented by Cardoso et al. [16], which indicated interactions of LIN with geraniol and fluconazole (FLC) toward both *C. albicans* and *C. neoformans* isolates. The synergy was showed in the combination of LIN with geraniol (FICI = 0.284–0.38 for *C. albicans* strains). Their MIC values were reduced 4–32 times. In the case of the combination of LIN with FLC, two types of interactions were demonstrated: synergism (FICI = 0.134) and addition (FICI = 0.57) towards these strains. Their MIC values were reduced from 500 μg/mL and 0.975 μg/mL to 2.02 μg/mL and 0.065 μg/mL, respectively. The synergistic effect of LIN with geraniol (FICI = 0.3905) against *C. neoformans* was also observed. In the case of LIN in combination with FLC, their MIC values were also 4-fold reduced and addition was shown (FICI = 0.5077). It is worth noting that the concentrations of the two components necessary to eliminate these fungi together were very low [16].

Studies involving the assessment of the interaction of LIN with FLC were also conducted by other authors. In accordance with the results of Zore et. al. [50], the combination of LIN with this azole showed that terpenes exhibit excellent synergistic activity against *C. albicans* with FICI = 0.140. The authors suggested that LIN (at 0.008% (*v*/*v*) concentration) could reduce the MIC of FLZ by 64-folds (from 64 μg/mL to 1 μg/mL). Moreover, similar studies were conducted by Pandurang et al. [9]. These research works also showed a synergistic effect for the combination of LIN with FLZ. LIN acted with this azole against planktonic as well as developing biofilm forms of reference *C. albicans* ATCC 90028 strain with FICI = 0.156 and FICI = 0.3, respectively. The MIC values of both components decreased significantly. In turn, in relation to the clinical isolate of *C. albicans,* addition (FICI = 0.6) or synergism (FICI = 0.312) was indicated for these forms, respectively.

The next data [28] presented the effect of a combination of LIN with FLC against isolates of *C. krusei.* FICI values of terpene with azole were in the range 0.19–0.63. The addition and synergy were exhibited in twelve (57.1%) and nine (42.9%) isolates. LIN decreased MIC values of FLZ from 74.66 μg/mL even to 9.81 μg/mL in the case of these strains. In turn, other studies [70] showed that LIN and α-longipinene synergistically reduce the biofilm formation.

To sum up the results of our own research and that of other authors, the combination of LIN with selected compounds can significantly increase their antifungal effectiveness using their synergistic and additive effects. LIN in the right concentration and combinations may be useful to control fungal infections and can provide an alternative or complementary therapy in the treatment of candidiasis [71]. These results may suggest the possibility of including LIN as an ingredient in several mouthwash and gargle products used to relieve the symptoms of sore throats, sensitive gums, and mouth ulcers, as well as to eliminate fungal infections [18]. Moreover, due to its protective effects and low toxicity, LIN can be used as an adjuvant of antifungals or antiseptics. Therefore, it has a great potential to be applied as a natural and safe alternative form of therapy [71]. The antifungal properties of the mixture of LIN with selected antifungals may be used for creating new products.

These findings are very promising for the development of new therapeutic options for candidal infections.

## 5. Conclusions

The findings reported in this research showed some susceptibility in vitro of the reference strains of yeast belonging to selected species from *Candida* spp., *Cryptococcus* spp., *Geotrichum* spp., and *Saccharomyces* spp. to LIN. Additionally, the potential activity of LIN against *Candida* spp. isolates from clinical patients was confirmed. Evaluation of the terpene's mechanism of action showed that it can bind to ergosterol in the cell membrane, causing cell death. Moreover, a stronger anticandidal effect of LIN can be obtained by combining it with other antimycotic agents, especially with cetylpyridinium, chlorhexidine, and triclosan, due to promising synergistic interactions. It can enhance their efficiency and find wide application in the treatment of superficial fungal infections, especially oral candidiasis. Synergistic interactions between LIN and some antiseptics may provide the basis for the development of new antifungal formulations. However, this requires further clinical studies.

**Author Contributions:** Conceptualization, A.B.; methodology, A.B.; formal analysis, A.B.; investigation, A.B.; data curation, A.B.; writing, A.B.; writing—review and editing, A.B.; supervision, A.M. and A.B.; project administration, A.M. and A.B.; funding acquisition, A.M. All authors have read and agreed to the published version of the manuscript.

**Funding:** This research was funded by the Medical University of Lublin (DS 30).

**Institutional Review Board Statement:** The study was conducted in accordance with the Declaration of Helsinki and approved by the Ethical Committee of the Medical University of Lublin (No. KE-0254/75/2011).

**Informed Consent Statement:** Informed consent was obtained from all subjects involved in the study.

**Data Availability Statement:** The data presented in this study are available upon request from the corresponding author. The data are not publicly available due to privacy restrictions.

**Conflicts of Interest:** The authors declare no conflict of interest.

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
