# Peer review of "Synergistic Interactions between Linalool and Some Antimycotic Agents against Candida spp. as a Basis for Developing New Antifungal Preparations"

_applsci, doi:10.3390/app13095686_

Round 1
Reviewer 1 Report
Comments for applsci-2294462
This article is somewhat innovative and important for controlling fungal infections. However, there are still some small problems that need to be further modified in the text.
1. Line 59,The horizontal lines in "geraniol" should be deleted.
2. Line 65,"(C10H18O)" should be followed with a comma.
3. Line 77,“LINexhibit”should be two words。
4. Line 92,More comma in the “investigated.. In".
5. Line 144,More comma in the“standard. . Next”
6. Line 127,Fungi were grown at 35 ℃,but in the 2.2.1 section, the temperature was 37℃. Why?
7. Line 168,“incubated” in the “the microplates were incubated at 37°C” should be “placed”。
8. Line 173,“To evaluate if LIN bind to the fungal membrane sterols” is not smooth。
9. Line 230,The expression mode of “0.5 mg/mL to 8 mg/mL” is consistent with the previous, it should change to“0.5-8 mg/mL”。
10. In the Methods section, the calculation methods of MIC50, MIC 90, MFC 50, and MFC 90 are not introduced.
11. In Table 2 and Figure 1, the number of isolates did not correspond with the data in the material section.
12. Line 260,“protectstheir”should be divided into two words.
13. Line 264 and 275, Figure 1 should be Figure 2.
14. Line 308-309 and line 315-316, 3.91 , 7.81, 0.98 and 7.81 should add µg/mL.
15. In figure 3, (a) and (b) is different from the text description.
16. There are many kinds of units in the article, such as mM, M, mg/mL, µg/ mL, µl/ mL. The whole text should be unified.
17. Line 410,“0.5 to 1% shuld be “0.5% to 1%”。line414,“0.12–0.25%”should be “0.12%–0.25%”。
18. Line 416-440,Font is different from that found elsewhere.
19. Line 504,”(“ in “(of”' should be deleted.
20. Line 534,More comma in the “indifference..”.
21. Some of the literature is not followed by a DOI. Different presentation formats are also used for the DOI.
22. In some places, the language still needs to be reprocessed. Simplify the language even further.

Author Response
Dear Reviewer 1
Thank you very much indeed for your kind comments and the time spent on the revision of our manuscript. Here below we present a point-by-point list of the responses to the obtained comments.
This article is somewhat innovative and important for controlling fungal infections. However, there are still some small problems that need to be further modified in the text.
- Line 59,The horizontal lines in "geraniol" should be deleted.
Response: Thank you for this comment. The correction was introduced to the manuscript.
- Line 65,"(C10H18O)" should be followed with a comma.
Response: Thank you for this comment. The correction was introduced to the manuscript.
- Line 77,“LINexhibit”should be two words。
Response: Thank you for this comment. The correction was introduced to the manuscript.
- Line 92,More comma in the “. In".
Response: Thank you for this comment. The correction was introduced to the manuscript.
- Line 144,More comma in the“ . Next”
Response: Thank you for this comment. The correction was introduced to the manuscript.
- Line 127,Fungi were grown at 35 ℃, butin the 2.2.1 section, the temperature was 37℃. Why?
Response: Thank you for this comment. The correction was introduced to the manuscript.
The fungi in our research were incubated at 37°C. The temperature was standardized in all sections.
- Line 168,“incubated” in the “the microplates were incubated at 37°C” should be “placed”。
Response: Thank you for this comment. The correction was introduced to the manuscript.
- Line 173,“To evaluate if LIN bind to the fungal membrane sterols” is not smooth。
Response: Thank you for this comment. The correction was introduced to the manuscript.
We hope that the following sentence better shows the purpose of the study: „Another study was the evaluation of LIN binding to fungal membrane sterols using the exogenous ergosterol assay.”
- Line 230,The expression mode of “0.5 mg/mL to 8 mg/mL” is consistent with the previous, it should change to“0.5-8 mg/mL”。
Response: Thank you for this comment. The correction was introduced to the manuscript.
- In the Methods section, the calculation methods of MIC50, MIC90, MFC50, and MFC90 are not introduced.
Response: Thank you for this comment. The correction was introduced to the manuscript.
MIC50 and MIC90 values as well as the range of values obtained are important parameters for reporting results of susceptibility testing when multiple isolates of a given species are tested. The MIC50 represents the MIC value at which ≥50% of the isolates in a test population are inhibited; it is equivalent to the median MIC value. Given n test strains and the values y1, y2 … yn representing a graded series of MICs starting with the lowest value, the MIC50 is the value at position n × 0.5, as long as n is an even number of test strains. If n is an odd number of test strains, the value at position (n + 1) × 0.5 represents the MIC50 value. The MIC90 represents the MIC value at which ≥90% of the strains within a test population are inhibited; the 90th percentile. The MIC90 is calculated accordingly, using n × 0.9. If the resulting number is an integer, this number represents the MIC90; if the resulting number is not an integer, the next integer following the respective value represents the MIC90. A similar calculation applies to the MFC50 and MFC90 values [Schwarz et al., 2010].
[Schwarz, S.; Silley P.; Simjee, S.; Woodford, N.; van Duijkeren, E.; Johnson, A.P.; Gaastra, W. Editorial: assessing the antimicrobial susceptibility of bacteria obtained from animals. J. Antimicrob. Chemother. 2010, 65 (4), 601-604. doi: 10.1093/jac/dkq037]
- In Table 2 and Figure 1, the number of isolates did not correspond with the data in the material section.
Response: Thank you for this comment. In Table 2 and Figure 1, the number of isolates corresponds to the data in the material section. We hope to show below that these numbers match.
- Materials and Methods:
2.1.2. Microorganisms
„Additionally, 40 clinical isolates of Candida spp.: C. albicans (20 isolates) and non-albicans Candida spp. (NAC) (20 isolates: C. glabrata, C. tropicalis, C. parapsilosis, C. famata, C. krusei, C. lusitaniae, and C. guilliermondii) were used.”
- Results
3.1. The Antifungal Activity Assessment of LIN
Table 2:
Number of clinical isolates of C. albicans:
C. albicans = 8 (40%) isolates + 9 (45%) isolates + 3 (15%) isolates = 20 (100%) isolates
Number of clinical isolates of NAC:
NAC = 2 (10%) isolates + 11 (55%) isolates + 7 (35%) isolates = 20 (100%) isolates
Figure 1:
- at MIC value: C. albicans = 2 (10%) isolates + 3 (15%) isolates + 7 (35%) isolates + 5 (25%) isolates + 3 (15%) isolates + 0 (0%) isolates = 20 (100%) isolates
- at MFC value: C. albicans = 0 (0%) isolates + 3 (15%) isolates + 4 (20%) isolates + 5 (25%) isolates + 7 (35%) isolates + 1 (5%) isolates = 20 (100%) isolates
- at MIC value: NAC = 1 (5%) isolates + 5 (25%) isolates + 6 (30%) isolates + 6 (30%) isolates + 2 (10%) isolates + 0 (0%) isolates = 20 (100%) isolates
- at MFC value: NAC = 0 (0%) isolates + 2 (10%) isolates + 5 (25%) isolates + 4 (20%) isolates + 7 (35%) isolates + 2 (10%) isolates = 20 (100%) isolates
- Line 260,“protectstheir”should be divided into two words.
Response: Thank you for this comment. The correction was introduced to the manuscript.
- Line 264 and 275, Figure 1 should be Figure 2.
Response: Thank you for this comment. The correction was introduced to the manuscript.
- Line 308-309 and line 315-316, 3.91 , 7.81, 0.98 and 7.81 should add µg/mL.
Response: Thank you for this comment. The correction was introduced to the manuscript.
- In figure 3, (a) and (b) is different from the text description.
Response: Thank you for this comment. The correction was introduced to the manuscript.
- There are many kinds of units in the article, such as mM, M, mg/mL, µg/ mL, µl/ mL. The whole text should be unified.
Response: Thank you for this comment. We fully agree with this suggestion. According to EUCAST guidelines: „Results are recorded as … the Minimum Inhibitory Concentration (MIC), expressed in mg/L or μg/mL.” Therefore, we used the following units: mg/mL and μg/mL for high and low concentration values, respectively. Our results show the antifungal activity of LIN in mg/mL and the studied antifungals in μg/mL. For high MIC and MFC values, it is more convenient to use the mg/mL unit. The result is then more readable. However, some authors reported the antifungal activity of LIN (both MIC and/or MFC values) also in various types of units, such as µM, mM, M, µl/mL, % (v/v). We presented their data in our discussion. Therefore, we would like to request that the original MIC and MFC units of LIN of these researchers be retained.
- Line 410,“0.5 to 1% shuld be “0.5% to 1%”。line414,“0.12–0.25%”should be “0.12%–0.25%”。
Response: Thank you for this comment. The correction was introduced to the manuscript.
- Line 416-440,Font is different from that found elsewhere.
Response: Thank you for this comment. The correction was introduced to the manuscript.
- Line 504,”(“ in “(of”' should be deleted.
Response: Thank you for this comment. The correction was introduced to the manuscript.
- Line 534,More comma in the “indifference..”.
Response: Thank you for this comment. The correction was introduced to the manuscript.
- Some of the literature is not followed by a DOI. Different presentation formats are also used for the DOI.
Response: Thank you for this comment. The correction was introduced to the manuscript. The DOI were added and corrected. Only 4 references below do not have DOI, so websites are included:
Pandurang, M.S.; Devrao, H.S.; Ganpatrao, B.R.; Mohan, K.S. Lemongrass oil components synergistically activates fluconazole against biofilm forms of Candida albicans. J. Bacteriol. Mycol. 2018, 5 (3), 1069. https//austinpublishinggroup.com/bacteriology/fulltext/bacteriology-v5-id1069.php
European Committee for Antimicrobial Susceptibility Testing (EUCAST). Determination of minimum inhibitory concentrations(MICs) of antibacterial agents by broth dilution. EUCAST discussion document E. Dis 5.1 Clin. Microbiol. Infect. 2003, 9, 1–7. https://clinicalmicrobiologyandinfection.com/article/S1198-743X(14)64063-5/pdf
CLSI. Reference Method for Broth Dilution Antifungal Susceptibility Testing of Yeasts; M27-S4; Clinical and Laboratory Standards Institute: Wayne, PA, USA, 2017. https://clsi.org/media/1897/m27ed4_sample.pdf
Carrillo-Muñoz, A.J.; Giusiano, G.; Ezkurra, P.A.; Quindós, G. Antifungal agents: mode of action in yeast cells. Rev. Esp. Quimioter. 2006, 19 (2), 130–129. www.seq.es/seq/0214-3429/19/2/130.pdf
- In some places, the language still needs to be reprocessed. Simplify the language even further.
Response: Thank you for this comment. The correction was introduced to the manuscript.
We hope that the language has been properly processed and simplified more.
On behalf of the Authors,
Anna Biernasiuk

Reviewer 2 Report
The current study entitled "Synergistic Interactions between Linalool and Some Antimycotic Agents against Candida spp. as a Basis for Developing New Antifungal Preparations" shows antifungal activity of Linalool alone or in combination with other standard antimicrobials. The experiments are performed very well and the findings are interesting. However, the authors should revise the manuscript in the light of the following comments:
Abstract: Provide some quantitative findings in the abstract section.
Introduction: Give information about more phytochemicals and their antifungal activities in the introduction section.
Figure 2: Show units on Y-axis in micrograms per ml or miligram per ml.
Table 3. Statistical analyses should be performed to compare Linalool alone or in combination with standard drugs.
Author Response
Dear Reviewer 2
Thank you very much indeed for your kind comments and the time spent on the revision of our manuscript. Here below we present a point-by-point list of the responses to the obtained comments.
The current study entitled "Synergistic Interactions between Linalool and Some Antimycotic Agents against Candida spp. as a Basis for Developing New Antifungal Preparations" shows antifungal activity of Linalool alone or in combination with other standard antimicrobials. The experiments are performed very well and the findings are interesting. However, the authors should revise the manuscript in the light of the following comments:
Abstract: Provide some quantitative findings in the abstract section.
Response: Thank you for this comment. The correction was introduced to the manuscript. Some quantitative results were given in the abstract section.
Introduction: Give information about more phytochemicals and their antifungal activities in the introduction section.
Response: Thank you for this comment. The correction was introduced to the manuscript. Information on other phytochemicals and their antifungal activity towards Candida spp. was added in the introduction section.
Figure 2: Show units on Y-axis in micrograms per ml or miligram per ml.
Response: Thank you for this comment. Results presented in Figure 2 shows „The increase in the MIC values” (x MIC) of LIN and NYS (as control drug) in the presence of ergosterol and sorbitol against selected reference Candida spp. strains. The antifungal activity of LIN as MIC in units of µg/mL or mg/mL is presented in Tables 1-3, Figures 1 and 3. In turn, Figure 2 shows how the MIC values of both LIN and NYS changed in the presence of ergosterol or sorbitol compared to their MIC values without ergosterol or sorbitol. This Figure shows how many times the MIC value increased as „x MIC”. Therefore, we are unable to show the units here.
Table 3. Statistical analyses should be performed to compare Linalool alone or in combination with standard drugs.
Response: Thank you for this comment. The correction was introduced to the manuscript. The statistical analysis, using the the Mann-Whitney U test, was performed to compare the activity of LIN alone and in combination with antifungals (Table 3). The statistically significant differences in MIC values were observed for the synergistic combinations LIN and cetylpyridinium, LIN and chlorhexidine, LIN and triclosan (p = 0.016 or p = 0.012, respectively). In turn, in other combinations, a statistically significant difference was indicated only for LIN (p = 0.016) in combination with silver nitrate and LIN (p = 0.012) with nystatin.
On behalf of the Authors,
Anna Biernasiuk

Round 2
Reviewer 2 Report
The authors need to cite an important reference showing the use of a combination of Ginger extract and fluconazole against drug-resistant C. albicans.
Coadministration of Ginger Extract and Fluconazole Shows a Synergistic Effect in the Treatment of Drug-Resistant Vulvovaginal Candidiasis. Infect Drug Resist. 2021 Apr 21;14:1585-1599.
Author Response
Dear Reviewer 2
Thank you very much for this important comment and the time spent on the revision of our manuscript.
The authors need to cite an important reference showing the use of a combination of Ginger extract and fluconazole against drug-resistant C. albicans.
Coadministration of Ginger Extract and Fluconazole Shows a Synergistic Effect in the Treatment of Drug-Resistant Vulvovaginal Candidiasis. Infect Drug Resist. 2021 Apr 21;14:1585-1599.
Response: Thank you for this significant comment. The correction was introduced to the manuscript. The above reference showing the use of a combination of Ginger extract and fluconazole towards drug-resistant C. albicans was cited by us in the introduction section of our manuscript (References – no. 15).
Khan, A.; Azam, M.; Allemailem, K.S.; Alrumaihi, F.; Almatroudi, A.; Alhumaydhi, F.A.; Ahmad, H.I.; Khan, M.U.; Khan, M.A. Coadministration of ginger extract and fluconazole shows a synergistic effect in the treatment of drug-resistant vulvovaginal candidiasis. Infect Drug Resist 2021, 14, 1585-1599. doi: 10.2147/IDR.S305503
On behalf of the Authors,
Anna Biernasiuk
